# Pregnancy and Lactation-Associated Osteoporosis: Combined Pharmacological and Rehabilitative Management

**DOI:** 10.3390/jfmk10030336

**Published:** 2025-08-31

**Authors:** Rossana Gnasso, Ayda Tavakkolifar, Andrea Esposito, Antonella Malinconico, Giuseppe Esposito, Lucia Taddeo, Stefano Palermi, Alessandro Nunzio Velotti, Antonio Picone, Carlo Ruosi

**Affiliations:** Department of Physical Medicine and Rehabilitation, Federico II University Naples, Via Pansini 5, 80131 Naples, Italy; tavakolifarayda@gmail.com (A.T.); andreaespo93@gmail.com (A.E.); antonella.malinconicoam@gmail.com (A.M.); esposito.giuseppe88@gmail.com (G.E.); luciataddeo1990@gmail.com (L.T.); stefano.palermi@unina.it (S.P.); alevelo94@gmail.com (A.N.V.); antonio.picone@unina.it (A.P.); caruosi@unina.it (C.R.)

**Keywords:** osteoporosis, pregnancy, calcium, rehabilitation

## Abstract

**Background**: Pregnancy and lactation-associated osteoporosis is a rare cause of severe skeletal fragility in young women typically presenting with vertebral compression fractures during late pregnancy or postpartum. Its rarity and lack of risk factors often delay diagnosis. **Case presentation**: The patient was a 34-year-old pregnant Italian woman, presenting with severe osteoporosis related to pregnancy and lactation. The patient presented for the first time at the Outpatient clinic of the Rehabilitation Unit in the Department of Public Health at the University of Federico II, Naples in March 2024, exhibiting severe symptoms indicative of osteoporosis, along with acute lower back pain. During the anamnesis, it was revealed that the patient was unable to bend forward, with reduced flexion and extension movements. The symptoms began during the third trimester. **Management and diagnosis**: In terms of diagnosis, clinical exams were conducted to confirm the disease. The MRI exam showed fractures and vertebral variations, with significant findings including calcification. Additionally, DXA indicated lower values compared to normal Treatment included: breastfeeding cessation, correction of calcium and vitamin D deficiencies, and bisphosphonate injection therapy. It is noteworthy that the rehabilitative approach has been recommended throughout pharmacological treatment and especially upon its suspension. Ultimately, the primary cause of this condition was pregnancy as bone resorption increases during pregnancy. **Outcome**: Following clodronate treatment completion, the patient showed full clinical recovery and significant radiological improvement. Follow-up DXA one year after diagnosis revealed normalized bone density and the patient had gained autonomy in activities of daily living with no further symptoms.

## 1. Introduction

Osteoporosis is a significant public health issue affecting millions worldwide. The disease can be classified into primary and secondary forms based on underlying etiology.

Primary Osteoporosis (85–90% of cases)

Postmenopausal (Type I): The most common type of osteoporosis, affecting 1 in 3 women over 50. Prevalence increases dramatically with age from 14% (ages 50–59) to 70% (age 80+).

Senile (Type II): This type affects both genders over 70 years, with 1 in 5 men over 50 affected. Juvenile/Idiopathic: Rare form occurring in children and young adults.

Secondary Osteoporosis (10–15% of cases)

Secondary osteoporosis represents a significant clinical concern, affecting two-thirds of older men, more than 50% of premenopausal women, and 30% of postmenopausal women. Notably, over 50% of premenopausal women with secondary osteoporosis present with associated articular and muscle-tendon disorders [1,2]. Common secondary causes include glucocorticoid-induced osteoporosis (most frequent), endocrine disorders, malabsorption syndromes, chronic kidney disease, and malignancy-related bone disease. Regarding the overall distribution, the male-to-female ratio remains approximately 1:4. While postmenopausal osteoporosis predominates as the most common type globally, secondary osteoporosis shows higher relative frequency in men and premenopausal women, with the latter group frequently presenting concurrent musculoskeletal pathology [3]. In Italy, about 5 million people suffer from osteoporosis, of which approximately 2.5 million are women. Italian women demonstrate the Mediterranean Paradox with vitamin D deficiency affecting 30–60% of Southern European populations compared to <20% in Northern Europe, despite Italy’s geographic solar advantage. Winter complications are particularly pronounced in Italian populations, with mean serum 25(OH)D levels falling below sufficient values by winter’s end and Italian children showing significantly lower winter levels (21.6 ng/mL) versus summer (46.7 ng/mL). Conversely, Nordic countries like Sweden maintain adequate levels (mean 71.0 nmol/L) with 79.2% achieving adequate status through systematic dietary fortification strategies. Ethnic variations show White women in England maintaining higher winter levels (39.9 nmol/L) compared to Asian women (16.9 nmol/L) in the same location, while African American groups demonstrate lower 25(OH)D levels than non-Hispanic white groups but paradoxically maintain better bone mineral density. Public health approaches vary dramatically with supplement usage ranging from 6.7% in Greek women to 65.8% in Danish women, where Nordic countries compensate for geographic disadvantages through mandatory food fortification while Mediterranean countries inadequately rely on assumed solar synthesis. These data indicate Italian women face greater winter vitamin D complications than Northern European counterparts despite geographic advantages, necessitating targeted supplementation strategies particularly during reproductive years when deficiency can contribute to pregnancy-related osteoporosis [4,5]. Pregnancy and lactation-associated osteoporosis (PLO) is a very rare type of osteoporosis that occurs in the third term of pregnancy or after delivery. It happens approximately 4–8 times per million deliveries. It was mentioned for the first time by Nordin Roper in 1955. There are around 100 documented cases in the world. It is a rare and often underestimated form of skeletal fragility that affects women during pregnancy and postpartum. PLO typically presents as thoracolumbar back pain and can lead to vertebral compression fractures (VCFs). Therefore, PLO should be considered in the differential diagnosis when a woman experiences debilitating back pain during pregnancy or while breastfeeding [6].

The management of PLO remains controversial, but it is crucial to achieve an early diagnosis, consider lactation interruption, and apply anti-osteoporosis therapy along with regular follow-ups. These measures are essential for prevention and treatment, significantly improving the mother’s quality of life during her activities of daily living.

In this article, a rare case of a 34-year-old Italian woman diagnosed with PLO during late pregnancy and early lactation after the birth of her child was discussed. This case is particularly interesting since osteoporosis typically occurs at older ages or in individuals with medical or familial histories. As a result, this case was studied further and the investigated findings were depicted. The aim of study was to highlight the diagnostic approach, treatment rationale and rehabilitation strategies applicable in similar cases.

## 2. Materials and Methods

### 2.1. Patient History and Presentation

This case report, written in accordance with the CARE guidelines, describes the diagnostic and rehabilitation process of a 34-year-old Caucasian female, born and raised in Italy, with a sedentary lifestyle and employed in office work. The patient had no significant medical comorbidities and maintained excellent general health until the third trimester of pregnancy, during which she developed a disabling osteoarticular disorder.

In March 2024, a 34-year-old woman, gravida 1, para 1, presented to the Outpatient Rehabilitation Clinic at the Department of Public Health, University of Naples Federico II. She reported a three-month history of thoracolumbar and lumbosacral pain with onset during the third trimester of pregnancy, persisting into the postpartum period. There was no history of antecedent trauma.

The patient described mechanical-type pain with radiation to the lumbar region and bilateral gluteal areas. She experienced acute exacerbations of back pain, with significant functional limitation affecting spinal range of motion, particularly restricting lumbar flexion, forward flexion, and extension.

### 2.2. Obstetric and Medical Background

The patient’s medical history did not indicate any connection to osteoporosis. The patient was suffering from gastroesophageal reflux both before and during her pregnancy. Prior to her pregnancy, she had normal levels of calcium and vitamin D, as well as normal bone densitometry results. Furthermore, she had no history of surgical interventions, systemic and/or genetic disorders. Her pregnancy proceeded straightforward. The patient did not smoke, and her body mass index (BMI) was 24. She was employed and did not typically experience any difficulties with her activities of daily living (ADL). She maintained a regular nutritional status both before and during her pregnancy. Additionally, her family history did not reveal any cases of osteoporosis.

### 2.3. Initial Imaging and Referral

The patient delivered without complications in December 2023. The lower back pain persisted even after delivery. In January 2024, the gynecologist conducted a postpartum follow-up examination due to the patient’s presentation of multiple symptoms, including severe pain. Subsequently, magnetic resonance imaging (MRI) was ordered for further evaluation. The results indicated very slight deformities in the T7, T9, and T12 vertebral bodies due to vertebral fractures located in the disco-somatic regions. There was also a linear edematous band present in the vertebral bodies, along with some depression in the superior limits of T7 and T9, and a slight depression in the inferior limit of T12. No additional pathologies, such as herniations or disc issues, were observed in this MRI (Figure 1). Consequently, the gynecologist referred the patient to the aforementioned Rehabilitation Department. It is important to note that, based on the presence of fractures, the gynecologist recommended avoiding breastfeeding to prevent further worsening of the condition; however, the patient refused to do so. Due to persistent and intensifying pain, the patient was referred to a physiatrist.

### 2.4. Diagnostic Workup

In January 2024, the patient presented to the clinic with her history and showing MRI results (Figure 1). Due to suspected PLO, Dual-energy X-ray Absorptiometry (DXA) was recommended. The results indicated a lumbar T-score of −3.4, which is indicative of osteoporosis (as values below −2.5 are considered below normal), and a femoral T-score of −1.3. Additionally, laboratory tests revealed low levels of calcium and vitamin D. Laboratory Workup for Osteoporosis consisted of the following steps:

Primary laboratory evaluation, including complete blood count, comprehensive metabolic panel, liver function tests, inflammatory markers (ESR, CRP), and thyroid function studies to exclude secondary causes.

Bone metabolism assessment comprising 25-hydroxyvitamin D, parathyroid hormone (PTH), and bone turnover markers including bone-specific alkaline phosphatase, osteocalcin, and C-terminal telopeptide (CTX).

Finally, secondary osteoporosis screening, including 24 h urinary free cortisol, sex hormones (testosterone in men, estradiol/FSH/LH in women), serum protein electrophoresis, anti-tissue transglutaminase antibodies and immunoglobulin levels [7].

Table 1 presents the laboratory results provided by the patient during the outpatient evaluation. Consequently, a diagnosis of PLO was made.

### 2.5. Treatment Strategies

A conservative therapeutic approach was undertaken, including both pharmacological and rehabilitative strategies

(a)Rehabilitation: primary activity restrictions included strict avoidance of heavy lifting and cessation of breastfeeding. Orthotic management consisted of a rigid three-point hyperextension thoracolumbosacral orthosis with pelvic band. The bracing protocol required eight hours of daily wear for three months, initiated immediately upon osteoporosis diagnosis. During the orthotic treatment period, patients were instructed to avoid excessive spinal movements including torsion, rotation, flexion and extension of the thoracolumbar spine.(b)Pharmacological: next, the pharmacological phase was initiated. Since the patient had osteoporosis along with severe lower back pain and a fracture, bisphosphonate was prescribed, which is the first-line treatment for osteoporosis. On the other hand, due to severe pain and bone edema in the vertebral bodies and heartburn that the patient was suffering from, clodronate was prescribed instead of an oral bisphosphonate, alendronate, as one of its main side effects is further heartburn and epigastric discomfort. The dosage used in this treatment program was 200 mg via intramuscular administration. The clodronate prescription was as follows: during the first month (March 2024), the drug was intramuscularly administered every day for the first 10 days. In the remaining 20 days, it was administered every 2 days. In the second month (April 2024), using one intramuscular vial was recommended each week for the next 30 days. In this month, the second MRI was repeated in order to see the follow-up results. In the third month (May), the drug was administered every two weeks intramuscularly. Supplemental therapy included calcium citrate (one stick daily for 30 days) and vitamin D (25,000 I.U. oral vials) administered weekly for the first 30 days, then biweekly for six months.

### 2.6. Follow-Up and Outcome

After initiating conservative treatment, the MRI after 90 days was carried out and the results showed a complete disappearance of the vertebral compression fractures. All previously identified fractures were stabilized and the previously noted edematous band in the spongy part of the vertebral body had disappeared (Figure 2).

The therapeutic regimen included continuation of pharmacological management with calcium and vitamin D supplementation and weekly intramuscular clodronate administration for six months. Activity restrictions regarding weight-bearing and lactation were maintained. Following the second MRI findings, the rehabilitation protocol was modified accordingly.

The orthotic weaning process involved gradual reduction in brace wearing time over a 30-day period. Concurrently, lower extremity strengthening exercises were initiated to maintain muscle strength. Upon brace discontinuation—coinciding with radiological resolution of vertebral fractures—progressive spinal stabilization exercises were implemented using controlled movement patterns. A comprehensive stretching program was incorporated into the treatment plan.

The physiotherapy approach utilized assisted mobilization techniques, combining passive range of motion performed by the therapist with patient-assisted active movements. The rehabilitation program consisted of conventional physiotherapy and aquatic therapy (hydrokinesitherapy), with each modality scheduled twice weekly for nine months.

## 3. Results

In January 2025, twelve months post-osteoporosis diagnosis, follow-up DXA was performed. The results demonstrated significant clinical improvement with resolution of osteoporosis: lumbar spine T-score improved to −2.3 and femoral neck T-score to −0.4.

Given the downgrading to mild osteopenia, clodronate therapy was discontinued and the patient was maintained on calcium and vitamin D3 supplementation as monotherapy. The rehabilitation maintenance program comprised low-intensity, prolonged aerobic exercise (60 min sessions, twice weekly for a minimum of eight weeks) to preserve therapeutic gains.

By January 2025, the patient demonstrated complete restoration of lumbar spine mobility with unrestricted lateral rotation and flexion–extension range of motion, reporting full functional independence in ADLs.

All procedures described in this case report adhered to institutional research committee ethical standards and the Declaration of Helsinki. Written informed consent for study publication was obtained from the patient.

The following timeline (Figure 3) illustrates the chronological sequence of diagnostic procedures and clinical assessments:

## 4. Discussion

### 4.1. Epidemiology and Clinical Presentation

Pregnancy and lactation-related osteoporosis is a rare condition affecting a small percentage of individuals, with limited research compared to general osteoporosis, making exact statistical data unavailable. The condition presents primarily as severe and disabling lumbar pain caused by multiple vertebral fractures, which can restrict spinal mobility, though femoral or pelvic pain may occur less commonly. Diagnosis remains primarily clinical due to the absence of official diagnostic criteria, requiring identification of osteoporotic features including spontaneous vertebral fractures and low bone mineral density during or after pregnancy, while systematically excluding secondary causes such as hematologic, endocrine, rheumatologic, renal, gastrointestinal disorders and drug-induced etiologies [8]. Case reports conducted by Jun Jie in 2020 highlighted that the pathophysiology of this disease remains unclear, but there are several theories regarding it. The primary theory suggests that it involves altered calcium metabolism with maternal–fetal calcium transfer during pregnancy compensated by increased intestinal absorption, while lactation relies on bone resorption through osteoclast activation via the OPG-RANK-RANKL pathway and parathyroid hormone-related peptide secreted by mammary tissue [9]. Although bone density loss is physiologically expected and typically reversible in normal pregnancies, significant risk factors include smoking, alcohol consumption, previous vertebral fractures, history of trauma or interventions, genetic predisposition, immobilization, low BMI, vitamin D deficiency and certain medications such as corticosteroids, heparin, and anticonvulsants, with pathological bone loss predominantly affecting the thoracolumbar spine. One potential complication suggested by Gehelen et al. is preterm birth, though this association requires further verification through additional research [10]. Among the risk factors listed above that predispose patients to PLO, the most frequent is vitamin D deficiency with high incidence even in the sunniest countries. As mentioned in the Introduction, Southern Italy exhibits significant seasonal vitamin D fluctuations (17.8% winter deficiency vs. 2.2% summer, with more pronounced variations in women at 27.8% vs. 3.4%) despite abundant solar radiation, due to solar protection use, inadequate dietary intake, and lack of food fortification—paradoxically showing higher deficiency rates than Nordic countries. This seasonal vulnerability, particularly affecting children during winter and spring, necessitates proactive supplementation strategies and monitoring in Southern Italian clinical practice, especially for PLO prevention where vitamin D deficiency during winter conception or lactation periods significantly increases risk, requiring targeted supplementation protocols to maintain optimal bone health throughout the reproductive cycle [11].

### 4.2. Treatment Strategies

Treatment goals should always focus on improving quality of life and relieving symptoms, such as lower back pain, while also preventing new vertebral fractures. The treatment plan should be personalized, considering each patient’s age, bone mineral density, and other individual factors. Although bisphosphonates have not yet been fully validated for certain conditions, recent trials suggest they may have positive effects on symptom relief. Li et al., in 2018, showed that either zolendronic acid and alendronate could reduce bone pain, improve β-CTX level and increase Bone Mineral Density (BMD) in 12 patients affected by PLO [12].

Among monoclonal antibodies with established antiresorptive properties, denosumab has demonstrated therapeutic efficacy in pregnancy PLO. Strumpf and colleagues reported clinically significant improvements in BMD in the lumbar spine, femoral neck, and total hip, with no subsequent fractures observed following 18 months of denosumab therapy [13].

Teriparatide, an anabolic agent with proven bone-forming properties, has also shown promise in PLO management. A recent systematic review by Ali et al. (2024) [14] analyzed the efficacy of teriparatide in this patient population. Despite the inherently elevated fracture risk associated with PLO, their analysis of 175 cases revealed that only 14.7% (20/175) of patients treated with teriparatide sustained new fractures during follow-up periods ranging from 9 months to 9 years. In addition, Cerit et al [15] reported a case of PLO associated with vertebral fractures treated through teriparatide, after the failure of conservative therapy (thoracolumbar orthoses, discontinuing lactation and supplementation of calcium and vitamin D). The authors report clinical benefits at 12 months with an increase in BMD on DXA. 

As a final note, a very recent case report conducted by Nomura in April 2025 [16] showed the benefits of romosozumab in a middle-aged patient affected by PLO associated with several thoracic vertebral and sacral fragility fractures. After one year of romosozumab therapy, her lumbar BMD increased, low back pain effectively ameliorated and no further fractures occurred.

### 4.3. Prognosis and Follow-Up

The prognosis of the disease largely depends on each patient’s medical condition and various other factors, such as early diagnosis, interventions, previous risk factors (including prior pelvic and vertebral fractures), and bone mineral density. Generally, the prognosis is very good if the diagnosis is made early and the woman adheres to the treatment program. One limitation of our study is the restricted patient cohort; however, it should be emphasized that, to our knowledge, this represents the first case report documenting the significant therapeutic benefits of clodronate treatment combined with comprehensive rehabilitation management in pregnancy and lactation-associated osteoporosis (PLO) in a woman of Italian descent.

## 5. Conclusions

Pregnancy-associated osteoporosis (PAO) represents an underdiagnosed condition affecting approximately 6.8 per 100,000 pregnancies, with 86% of cases occurring during first pregnancy. The condition typically presents with sudden onset of multiple vertebral compression fractures (4–10 vertebrae) during late pregnancy or lactation, causing severe lumbosacral pain and significant functional impairment. The pathophysiology involves altered bone remodeling rates, distinguishing it as a severe, early-onset osteoporosis in young women.

Management approaches include cessation of breastfeeding, orthopedic support, calcium and vitamin D supplementation, and targeted pharmacotherapy. However, optimal treatment protocols remain undefined due to limited evidence and the potential for spontaneous improvement.

The condition significantly impacts maternal quality of life, physical function, and psychological wellbeing, potentially affecting maternal–infant bonding.

Clinicians must maintain heightened clinical suspicion for pregnancy and lactation-associated osteoporosis (PLO) in any pregnant or postpartum woman presenting with persistent lumbosacral pain, regardless of the presence or absence of traditional risk factors.

Further large-scale prospective studies are urgently needed to establish standardized diagnostic criteria, optimal treatment protocols and long-term outcomes for this poorly understood but clinically significant condition in order to validate these therapeutic results.

## Figures and Tables

**Figure 1 jfmk-10-00336-f001:**
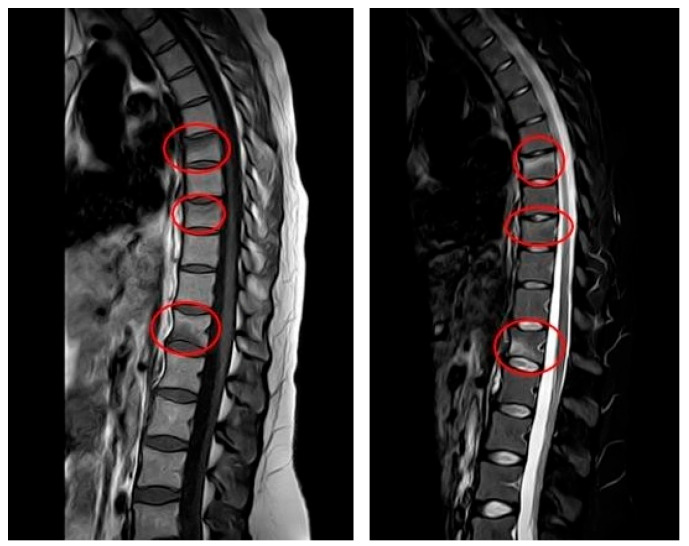
*MRI at initial diagnosis (January 2024, one month postpartum):* MRI (T1—weighted sequence on the (**left**), T2—weighted sequence on the (**right**)) showing anterior wedge compression fractures at T7, T9 and T12, with linear bone marrow edema and mild depression of vertebral endplates. No disc herniation or spinal canal narrowing is present.

**Figure 2 jfmk-10-00336-f002:**
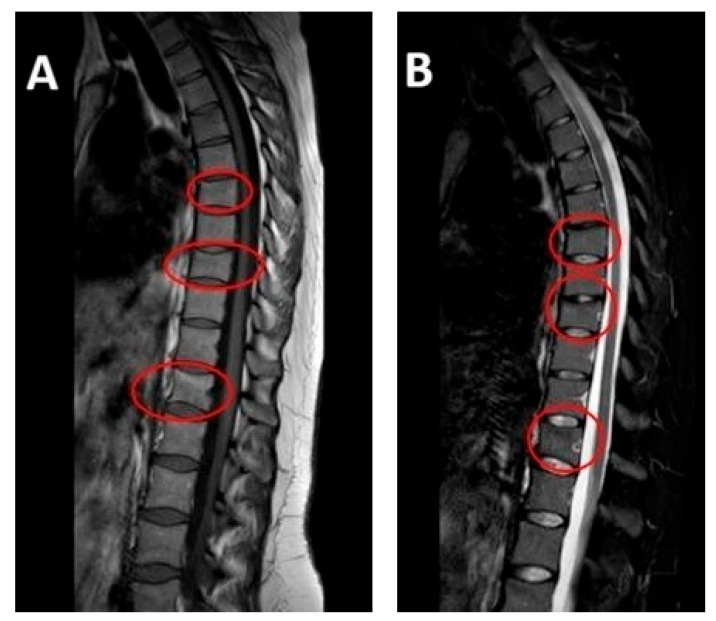
*MRI at follow-up (April 2024, at four-month postpartum follow-up*): repeat sagittal MRI showing resolution of edema and stabilization of previous vertebral compression fractures. Vertebral alignment is preserved with no evidence of new pathology. Subfigure (**A**) on the left shows T1-weighted sequence on the left. Subfigure (**B**) shows T2-weighted sequence on the right.

**Figure 3 jfmk-10-00336-f003:**
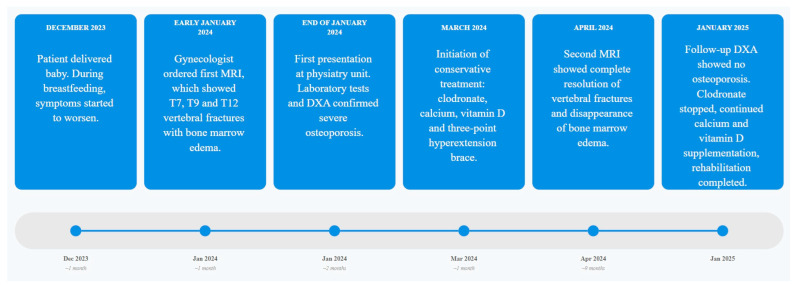
Chronological overview of patient’s clinical pathway with linear timeline from: symptom onset, imaging, pharmacological treatment, rehabilitation phases and follow-up DXA results over a 12-month period.

**Table 1 jfmk-10-00336-t001:** Laboratory results at time of diagnosis (January 2024, one month postpartum).

Parameter	Value	Normal Range
Hemoglobin	12.9 g/dL	12.3–15.3 g/dL
Hematocrit	38.5%	36–48%
RBCs	4.97 × 10^9^/mm^3^	4.0–5.5 × 10^9^/mm^3^
MCV	83 fL	80–100 fL
MCHC	32.8 g/dL	32–36 g/dL
MCH	25.3 pg	25–32 pg
WBC	6.59 × 10^3^/mm^3^	4–10 × 10^3^/mm^3^
Platelets	298,000/mm^3^	140,000–450,000/mm^3^
Creatinine	0.81 mg/dL	0.5–1.0 mg/dL
Calcium	9.0 mg/dL	8.9–10.1 mg/dL
Phosphorus	4.3 mg/dL	2.5–4.5 mg/dL
Total Protein	8.2 g/dL	6.0–8.3 mg/dL
Albumin	5 g/dL	3.5–5.5 g/dL
ALT	12 U/L	7–45 U/L
AST	19 U/L	8–43 U/L
Alkaline Phosphate	119 U/L	40–140 U/L
25 OH Vitamin D	18.4 ng/mL	30–100 ng/mL
PTH	27.7 pg/mL	10–65 pg/mL
TSH	0.87 µUI/mL	0.4–4.0 µUI/mL

## Data Availability

The original contributions presented in this study are included in the article. Further inquiries can be directed to the corresponding author.

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
