# Peer review of "Pregnancy and Lactation-Associated Osteoporosis: Combined Pharmacological and Rehabilitative Management"

_jfmk, 2025, doi:10.3390/jfmk10030336_

Round 1

Reviewer 1 Report

Comments and Suggestions for Authors

There is a well-documented case report of a 34-year-old female patient with prostate cancer who was treated with clodronate, calcium supplements, and vitamin D supplements, resulting in an increase in bone mineral density.

The following points require improvement:

Abstract:

L24. It is necessary to add that the patient weaned from breastfeeding.

L29. It should also be noted that clodronate administration was discontinued.

Case presentation

L82. Line breaks are inappropriate.

L90. Were tests performed to rule out Cushing's syndrome as a cause of secondary osteoporosis?

L100. A normal range should be added to Table 1.

L103. "a" to "A"

L113. Bisphosphonate is considered a first-line treatment for postmenopausal patients with osteoporosis. However, because teratogenicity must be considered when using Bisphosphonate in premenopausal women, it is necessary to inquire about the patient's desire to have a second child.

L138. European guidance lists Bisphosphonates, including alendronate, ibandronate, and risedronate, as first-line treatments. However, we believe that clodronate is not the first choice. The reasons for its use and the choice of drug should be described in detail.

L197. Treatment strategies should include teriparatide preparations and denosumab, so the differences in their use should also be described.

Author Response

Kind regards,

dr. Rossana Gnasso 

Reviewer 2 Report

Comments and Suggestions for Authors

The manuscript by Rossana Gnasso et al. is a case report. This type of papers is not present among the ones listed in instructions for authors, however there are a few case reports published in Journal of Functional Morphology and Kinesiology. Anyway, the title must be shortened. I also recommend to check examples of case reports in the journal.

All abbreviations should be introduced upon first use.

The sentence in lines 76-78 is complex and may cause misunderstandings. It should be divided and rephrased.

Line 82 - a typo (paragraph).

Line 87 - "As a result" - There is no evidence for such phrase.

Lines 90-92 - The sentence is too long and complex.
Figures and Tables should include an indication of the time after birth of the child.

Line 95 - indicate normal levels of Ca and vitamin D here or in the table.

Line 125 - "It is important to note that" should be substituted or excluded. The following sentence should be modified, as it is of no less importance than the previous ones. 
The Figure 3 should be improved. The font should be much larger. Time intervals should be indicated. A linear timeline should be added below. Small illustrative images may be added.

Line 216. - "relatively small sample" - please rephrase the sentence.

Discussion should better describe potential signs and possible markers, which could be predictors of pregnancy-related osteoporosis.

Additionally, indicate frequencies of different kinds of osteoporosis, when compared in the Introduction. Data for Italian women are of more interest for this paper, however a small comparison to other ethnicities and regions should be added especially regarding winter complications with the levels of vitamin D in Northern Europe and other regions.

A remark on seasonal changes in vitamin D levels (and its importance for Southern Italy) should be included in the Discussion too.

Description of Methods should be added as a special section as suggested in Instructions for authors. The methods should include also the description of laboratory tests with appropriate references.

Author Response

Kind regards,

dr. Rossana Gnasso

Round 2

Reviewer 2 Report

Comments and Suggestions for Authors

The authors have satisfactorily addressed my comments and suggestions. I only would like to point out that 1) the font size of figure 3 should be increased and 2) the contrast of the colors (light blue vs white and gray vs light gray) should be improved.